# Predicting Outcome in Patients with Brain Injury: Differences between Machine Learning versus Conventional Statistics

**DOI:** 10.3390/biomedicines10092267

**Published:** 2022-09-13

**Authors:** Antonio Cerasa, Gennaro Tartarisco, Roberta Bruschetta, Irene Ciancarelli, Giovanni Morone, Rocco Salvatore Calabrò, Giovanni Pioggia, Paolo Tonin, Marco Iosa

**Affiliations:** 1Institute for Biomedical Research and Innovation (IRIB), National Research Council of Italy, 98164 Messina, Italy; 2Pharmacotechnology Documentation and Transfer Unit, Preclinical and Translational Pharmacology, Department of Pharmacy, Health Science and Nutrition, University of Calabria, 87036 Rende, Italy; 3S. Anna Institute, 88900 Crotone, Italy; 4Department of Engineering, Università Campus Bio-Medico di Roma, Via Alvaro del Portillo 21, 00128 Rome, Italy; 5Department of Life, Health and Environmental Sciences, University of L’Aquila, 67100 L’Aquila, Italy; 6San Raffaele Sulmona Institute, 67039 Sulmona, Italy; 7IRCCS Centro Neurolesi “Bonino-Pulejo”, 98123 Messina, Italy; 8Department of Psychology, Sapienza University of Rome, 00185 Rome, Italy; 9Santa Lucia Foundation IRCSS, 00179 Rome, Italy

**Keywords:** machine learning, linear regression, brain injury, prediction model, stroke, traumatic brain injury

## Abstract

Defining reliable tools for early prediction of outcome is the main target for physicians to guide care decisions in patients with brain injury. The application of machine learning (ML) is rapidly increasing in this field of study, but with a poor translation to clinical practice. This is basically dependent on the uncertainty about the advantages of this novel technique with respect to traditional approaches. In this review we address the main differences between ML techniques and traditional statistics (such as logistic regression, LR) applied for predicting outcome in patients with stroke and traumatic brain injury (TBI). Thirteen papers directly addressing the different performance among ML and LR methods were included in this review. Basically, ML algorithms do not outperform traditional regression approaches for outcome prediction in brain injury. Better performance of specific ML algorithms (such as Artificial neural networks) was mainly described in the stroke domain, but the high heterogeneity in features extracted from low-dimensional clinical data reduces the enthusiasm for applying this powerful method in clinical practice. To better capture and predict the dynamic changes in patients with brain injury during intensive care courses ML algorithms should be extended to high-dimensional data extracted from neuroimaging (structural and fMRI), EEG and genetics.

## 1. Introduction

Brain injury consists of damage to the brain that is not hereditary, congenital, degenerative, or induced by birth trauma or perinatal complications. The injury results in a modification of the brain’s neural activity, structure, and functionality with a consequent loss of cognitive, behavioral, and motor functions. Head trauma, ischemic and hemorrhage stroke, infections, and brain tumors are among the most common causes of acquired brain injury.

Due to the severe social and economic burden of brain injury, the expectation of long-term outcome is an important factor in clinical practice, and this is particularly important after severe traumatic brain injury (TBI), which still presents high mortality and unfavorable outcome rates with a severe global disability [1].

Outcome prediction is basically determined by prior knowledge of patient physical factors, and mainly involves demographic factors as well as comorbidities that are equally important factors to evaluate for clinical and rehabilitative prognosis [2]. It is obvious that each different pathology determining a brain injury has different prognostic factors in relation to the specific deficits that characterize each clinical picture.

Concerning clinical and demographic outcomes in stroke patients, it has been demonstrated that younger age, no injured corticospinal tract, residual good leg strength, continence, the absence of unilateral spatial neglect or other cognitive impairment, the level of trunk control, and independence in activities of daily living in patients, predict independent walking 3 months later injury in patients who are non-ambulatory after stroke [3,4,5].

Regarding the prognosis of TBI patients, it is well known that these patients achieve greater functional and cognitive improvements with respect to patients with cerebrovascular and anoxic aetiologies [6]. In TBI patients the most important clinical factors affecting outcome are age, Glasgow Coma Scale (GCS) score, Coma Recovery Scale-revised (CRS-r), pupil response, Marshall Computer Tomography (CT) classification, and associated traumatic subarachnoid hemorrhage. According to the literature, there is strong evidence for the prognostic value of the GCS on admission to the hospital which means lower admission GCS is associated with worse outcomes; furthermore, the GCS score shows a clear linear relationship with mortality. The most common CT classification used in TBI is the Marshall classification according to which levels III and IV are especially related to mortality, while levels I or II are more frequently associated with a favorable outcome. Specific outcomes have been reported in relation to individual CT characteristics, midline shifts, and mass lesions. As regards neuroimaging studies with magnetic resonance imaging (MRI), it has been demonstrated that the presence of diffuse axonal injury on MRI in patients with TBI results in a higher chance of unfavorable outcomes [7]. Gender does not seem to determine a difference in the outcome, while concerning the level of education higher educational levels are weakly related to a better outcome. Other important prognostic factors include hypotension, hypoxia, glucose, coagulopathy, and hemoglobin. In particular, hyperglycaemia and coagulopathy are the major determinants of disability and death in TBI patients. Indeed, prothrombin time showed a positive linear relationship with the outcome, where increasing values are associated with poorer outcomes [8,9,10,11]. One of the most recent and promising research lines of prognostic factors of rehabilitative outcomes regards biological markers in patients with brain injury determined by stroke or traumatic brain injuries [12,13].

In stroke patients, a recent meta-analysis has identified c-reactive protein, albumin, copeptin, and D-dimer to be significantly associated with long-term outcomes after ischemic BI [14]. In TBI Patients, novel and emerging predictors include the genetic constitution, advanced magnetic resonance imaging, and biomarkers. In particular, increased levels of interleukin (IL)-6, IL-1, IL-8, IL-10, and tumor necrosis factor-alpha are associated with worse outcomes, concerning both morbidity and mortality [15,16].

Although over the past few years many prognostic factors have been identified, relationships among demographical, clinical, biological, and psychological factors and outcomes could be not linear and intertwined. For this reason, most conventional approaches may fail in revealing these complex relationships.

Consequentially, machine learning (ML) approaches have emerged as a more robust way to discriminate between various classes of potential prognostic factors useful for predicting outcomes in TBI and stroke patients. In this prospective study we sought to summarize, for the first time, the main findings emerging from this new field of study, discussing the differences between traditional statistical methods (for example, linear regression) and the modern ML approaches, and future opportunities to be translated into primary care practice. We only discussed results from studies where ML algorithms were compared against traditional statistical methods such as logistic regression (LR).

## 2. Machine Learning Methods

ML is a subfield of Artificial Intelligence, studying the ability of computers to automatically learn from experience and solve specific problems without being explicitly programmed for it. These learning systems can continuously self-improve their performance and increase their efficiency in extracting knowledge from experimental data and analytical observations. ML includes three main approaches that differ in learning technique, type of input data and outcome, and typology of the task to solve: Supervised Learning, Unsupervised Learning and Reinforcement Learning.

Supervised learning is the most common paradigm of ML, applied when input variables and output targets are available, and relevant for neurorehabilitation clinicians. This approach consists of algorithms that analyze the mapping function between “input” and “output” variables with the goal to learn how to predict a specified “output” given a set of “input” variables, also called “predictors”. Supervised Learning can be broadly divided into two main types:Classification: where the output variable is made up of a finite set of discrete categories that indicate the class labels of input data, and the goal is to predict class labels of new instances starting from a training set of observations with known class labels;Regression: where the output is a continuous variable, and the goal is to find the mathematical relationship between input variables and outcome with a reasonable level of approximation.

The unsupervised approach is characterized by unlabeled input data. The algorithm explores and models data inherent in structure and patterns without the guidance of a labeled training set. Typical applications of Unsupervised Learning are:Clustering (or unsupervised classification): with the aim to divide data so that similarity of instances of the same cluster is maximized and similarity of different clusters is minimized;Dimensionality Reduction: where input instances are projected into a new lower-dimensional space.

Reinforcement Learning has the goal to develop a system called agent that improves through interaction with the environment. In particular, at each iteration the agent receives a reward (or a penalty) based on its action, which is a measure of how much this activity is good for the desired goal. An exploratory trial-and-error approach is exploited to find actions that maximize the cumulative reward. Very common applications of reinforcement learning are in computer games and robotics.

Among the wide number of possible machine learning algorithms, there are some conventional techniques that are considered the gold standard for classification problems and that have been employed in the studies presented in this review:LR [17]: the simplest among classification techniques, it is mainly used for binary problems. Assuming linear decision boundaries, LR works by applying a logistic function in order to model a dichotomous variable of output:
$$Logistic\;Function=\frac{1}{1+e^{-x}}$$
where *x* is the input variable.

This oversimplified model allows low training time and the poor possibility of overfitting, but at the same time, it may carry to underfitting for complex datasets. For these reasons, LR is suitable for simple clinical datasets such as those related to patients with brain injuries. Ridge Regression and Lasso Regression are distinguished from Ordinary Least Squares Regression because of their intent to shrink predictors by imposing a penalty on the size of the coefficients. Therefore, they are particularly useful in the case of big data problems:Generalized Linear Models (GLM) [18] are an extension of linear models where data normality is no longer required because predictions distribution $\hat{y}$ is transformed into a linear combination of input variables *X* throughout the inverse link function *h*:


(1)
$$\hat{y}\left(w,X\right)=h\left(Xw\right)$$


Moreover, the unit deviance d of the productive exponential dispersion model (EDM) is used instead of the squared loss function;


Support Vector Machine (SVM) [19]: it applies a kernel function with the aim to map available data into a higher dimensional feature space where they can be easily separated by an optimal classification hyperplane.k-Nearest Neighbors (k-NN) [20]: it assigns the class of each instance computing the majority voting among its k nearest neighbors. This approach is very simple but requires some not trivial choices such as the number of k and the distance metric. Standardized Euclidean distance is one of the most used because neighbors are weighted by the inverse of their distance:
$$d\left(q,x_{i}\right)=\sqrt{\overset{n}{\underset{i=1}{\sum }}{\left(\frac{q-x_{i}}{\sigma_{i}}\right)}^{2}}$$
where *q* is the query instance, $x_{i}$ is the *i*-th observation of the sample and $\sigma_{i}$ is the standard deviation;Naïve Bayes (NB) [21]: based on the Bayes’ Theorem, it computes for each instance the class with the highest probability of applying density estimation and assuming independence of predictors;Decision Tree (DT) [22]: a tree-like model that works performing for each instance a sequence of cascading tests from the root node to the leaf node. Each internal node is a test on a specific variable, each branch descending from that node is one of the possible outcomes of the test, and each leaf node corresponds to a class label. In particular, at each node the function *Information Gain* is maximized to select the best split variable:
$$Gain\left(A\right)=I-I_{res}\left(A\right)$$
where *I* represents the information needed to classify the instance and it is given by the entropy measure:$$I=-\overset{}{\underset{c}{\sum }}p\left(c\right)log_{2}p\left(c\right)$$With *p*(*c*) equal to the proportion of examples of class *c*.And *I_res_* is the residual information needed after the selection of variable *A:*$$I_{res}=-\overset{}{\underset{v}{\sum }}p\left(v\right)\overset{}{\underset{c}{\sum }}p\left(c|v\right)log_{2}p\left(c|v\right)$$A common technique employed to enhance models’ robustness and generalizability is the ensemble method [23,24,25,26] that combines predictions of many base estimators. The aggregation can be done with the Bootstrap Aggregation technique (Bagging) applying the average among several trees trained on a subset of the original dataset (such as in the case of Random Forests (RF)) or with the Boosting technique applying the single estimators sequentially giving higher importance to samples that were incorrectly classified from previous trees (like in AdaBoost algorithm);Artificial Neural Networks (ANNs) [27]: are a group of machine learning algorithms inspired by the way the human brain performs a particular learning task. In particular, neural networks consist of simple computational units called neurons connected by links representing synapses, which are characterized by weights used to store information during the training phase. A standard NN architecture is composed of an input layer whose neurons represent input variables *{x_i_| x_1_, x_2_, …, x_m_*}, a certain number of hidden layers for intermediate calculations, and the output layer that converts received values in outputs. Each internal node transforms values from the previous layer using a weighted linear summation (*u = w_1_x_1_ + w_2_x_2_ + … + w_m_x_m_*), followed by a non-linear activation function (*y =*
*ϕ(u + b*)) such as step, sign, sigmoid or hyperbolic tan functions. The learning process is performed throughout the backpropagation algorithm that computes the error term from the output layer and then back propagates this term to previous layers updating weights. This process is repeated until a certain stop criterion, or a certain number of epochs, are reached.


## 3. Predicting Outcome: Conventional Statistics versus Machine Learning

The recent literature that incorporates ML in the neurorehabilitation field raises a natural question: what is the innovation compared with conventional statistical techniques such as linear or LR? From one side, traditional statistics have long been used for regression and classification tasks, can also determine a relationship between input and output, and have been used for classification tasks. Some other authors may even claim that both linear and LR are themselves ML techniques, although some important distinctions needed to be made between classical statistical learning and ML (Table 1).

Statistical methods are top-down approaches: it is assumed that we know the model from which the data have been generated (this is an underlying assumption of techniques like linear and LR), and then the unknown parameters of this model are estimated from the data. The potential drawback is that the link between input and output is user chosen and could result in a suboptimal prediction model if the actual input–output association is not well represented by the selected model. This may occur if a user chooses LR, even though the relationship between input and output is non-linear, or when many input variables are involved.

Otherwise, ML methods are bottom-up approaches. No particular model is assumed, but starting from a dataset an algorithm develops a model with a prediction as the main goal. Generally, the resulting models are complex, and some parameters cannot be directly estimated from the data. In this case, the common procedure is to choose the best parameters either from previous relevant studies or tuning them during the training in order to give the best prediction. ML algorithms can handle a larger number of variables with respect to traditional statistical methods, but also require larger sample sizes for predicting the outcome with greater accuracy.

A potential limit of ML is that the repetition of the analysis may lead to slightly different results. The reliability of classical statistics is mainly related to the sampling process, but the same data lead to the same results independently of the number of times in which the same analysis was applied. The uncertainty is simply related to the concept that the sample was randomly extracted by the population. Techniques such as split-half, parallel form or bootstrap analysis have been introduced to retest the reliability of results among different resampled data. In ML, there is often an over-dimensioned system that could provide the same level of accuracy in predicting the outcome in different ways, and it means associated different weights to each variable even when the same model is applied to the same data sample. In a recent study, the importance associated with factors influencing harmonic walking in patients with stroke was found to have a variability going from 6% (for the iliopsoas maximum force) up to 37% (for the patient’s gender) [28].

### 3.1. Conventional Statistics versus Machine Learning Methods in TBI Patients

In the last few years, seven papers have been published aimed at comparing the performance of the regression models with respect to ML in extracting the best clinical indicators of outcome in TBI patients (Table 2). Historically, Amorim and colleagues firstly applied ML approaches to 517 patients with various severity of TBI. A large amount of demographic (gender, age), clinical (pupil reactivity at admission, GCS, presence of hypoxia and hypotension, computed tomography findings, trauma severity score), and laboratory data were used as predictors. Using a mixed ML classification model, they found that the naive Bayes algorithm had the best predictive performance (90% accuracy), followed by a Bayesian generalized linear model (88% accuracy) when mortality was used as an outcome. The most important variables used by ML models for prediction were: (a) age; (b) Glasgow motor score; (c) prehospital GCS; and (d) GCS at admission. In this paper, linear regression analysis has been directly merged into the ML models in order to improve prediction performance. Following a similar multimodal approach where a series of ML algorithms were individually used and finally pooled together in an ensemble model to evaluate the performance with respect to the LR approach, our group demonstrated high but similar performance among methods [29]. Indeed, we found similar performance among LM (82%) and ML (85%) algorithms when two classes of outcome approach (Positive vs. Negative measures of the Glasgow Outcome Scale-Extended (GOS-e)) were used. Age, CRS-r, Early Rehabilitation Barthel Index (ERBI), and entry diagnosis were the best features for classification. Tunthanathip et al., evaluated the performance of several supervised algorithms (SVM, ANNs, RF, NB, k-NN) compared to LR in a wide population of pediatric TBI. With respect to other studies, the traditional binary LR was performed with a backward elimination procedure for extracting the best prognostic factors useful to classification (GCS, hypotension, pupillary light reflex, and sub-arachnoid hemorrhage). The authors found that the SVM was the best algorithm to predict outcomes (accuracy: 94%). Instead, Gravesteijn et al. [30] directly compared LR with respect to a series of ML algorithms (SVM, RF, gradient boosting machines, and ANNs) to predict outcomes in more than 11,000 TBI patients. All statistical methods showed the same performance in predicting mortality or unfavorable outcomes (ranging from 79% to 82%), where the RF algorithm was the worst. Similarly, Nourelahi et al. [31] described the same results by evaluating 2381 TBI patients. Despite the employment of the only SVM and RF for ML analysis, they reached an accuracy in post-trauma survival status prediction of 79%, where the best features extracted were Glasgow coma scale motor response, pupillary reactivity and age. Similarly, Eftekhar et al. [32] only used the Artificial neural networks (ANN) algorithm to evaluate the prediction performance with respect to the LR model. ANN was able to predict mortality of TBI patients in almost all patients (95% of accuracy), although this performance was lower than LR (96%). Finally, following a one-single ML approach, Chong et al., used Neural Network to evaluate the predictive accuracy of different clinical data (i.e., presence of seizure, confusion, clinical signs of skull fracture). Evaluating data from a very small sample of TBI patients they reported high but similar performance among LR and ML approaches (93% versus 98%), indicating as best features a list of never reported clinical variables.

### 3.2. Conventional Statistics versus Machine Learning Methods in Stroke Patients

As shown in Table 3, for patients with stroke six studies have been included in this review because they compare the results of ML algorithms (ANN) with those of conventional regression analysis. The total number of patients included in these studies was very high (5346), going from 33 patients up to 2522. There was a wide variety of investigated outcomes, ranging from a return to work to death. Even wider was the variety of the assessed independent variables. The accuracy of ANN ranged between 74% and 93.9%, greatly depending also on the chosen method of analysis. The accuracy of conventional regression analysis was generally lower, ranging from 40 to 85%. In five out of these six studies, the ANN resulted in a more accurate prediction than conventional regression [28,36,37,38,39,40]. The unique exception was the study conducted on a large sample (2522 patients) in which the accuracy of ANN was slightly inferior (74% vs. 76.6%) [39]. Conversely, wider differences in favor of ANN were found for the two studies with the smaller sample size having as an outcome the functional status of patients at discharge [28,36]. When different types of ANNs were compared the Deep Neural Network [38] and the k-Nearest Neighbors [40], more accurate performance was detected. The features extracted by models were widely variable among studies leading to very different results, with some prognostic factors already well known in the literature such as older age [37,39].

## 4. Discussion

Since the similarity in performance reported for LR and ML approaches and the large heterogeneity in best features extracted, the main conclusion of this review is that ML does not confer substantial advantages with respect to LR methodologies in predicting the outcome of TBI or stroke patients (Figure 1). Qualitative evaluation of results suggested a trend towards better performance of ML algorithms in the stroke patients with respect to LR. However, without a quantitative comparison (i.e., benchmark analysis) a definitive conclusion cannot be drawn.

Despite using similar means, the boundary between LR (statistical inference) and ML is subject to debate. The LR had the advantage of identifying relationships between prognostic factors associating each of them with an odds ratio, while the use of ML is limited by the difficulty of interpreting the model, often used like a ‘black box’ to obtain the best performance on a specific test set. The most important advantage of ML algorithms is their capacity to perform non-linear predictions of the outcome and that do not require statistical assumptions such as independence of observations and multicollinearity. However, this common high non-linearity of the classification problem implies that the direction of effect of each input cannot be easily recognized [41]. An issue poorly investigated is the repeatability of the results obtained with ML. The two most important psychometric properties of a test are validity and reliability. The high accuracy found in the above-reported studies could be seen as proof of the validity of the ML approach. However, most of these studies identified specific prognostic factors but did not test the reliability of these findings if the ML was repeatedly applied. A recent study conducted on prognostic factors related to walking ability in patients with stroke showed that variability in the weight of each factor among 10 applications of an ANN analysis ranged between 6 up to 37%. On the other hand, authors reported that the reliability was lower for the factors with reduced weight, and higher for the most important factors [28]. However, this study highlights the need to assess not only the accuracy and hence the validity of the ML algorithms, but also their reliability [28].

For stroke patients, the accuracy of ML algorithms in predicting the outcome ranged from 74 to 95%. It is often reported that ANN requires wide samples for achieving good accuracy, however, the two studies of the six reviewed with the smaller samples showed higher accuracy of ANN with respect to conventional regression analysis [28,36], probably suggesting that also (or even more) regression needs wide samples to obtain solid results. It is important to also cite some other studies not reported in Table 2, for example that reporting accuracy of ANN of 100% for Wolf Motor Function Test scores in George et al., [41] were very different from the 30% for the functional independence measures in Sale et al. [42]. However, in the latter study, the outcome score was accurately predicted at 84%. The ANN is the most common type of ML algorithm used in stroke, conceivably for its higher simplicity with respect to other types, followed by support vector machines and random forest algorithms. Some studies compared the performances of different algorithms. It should be noted that literature also reports some papers about the accuracy of ML not compared with those of conventional statistical methods. Oczkowski and Barreca reported an accuracy of 88% on 147 patients with stroke [43]; George et al. reported an accuracy of 100% [41]; and Sale et al. [42] on 55 patients with an accuracy of 84% for predicting the Barthel Index score and of 30% for Functional Independence Measure; Thakkar et al. [44] on 239 patients with an accuracy between 81 and 85%; Liu et al. [45] on 56 patients 88–94%; Billot et al. [46] on 55 patients with an accuracy between 84 and 93%; and the study by Xie et al. [47] on a wide sample of patients (*n* = 512) reporting an area under the curve of the ROC analysis of 0.75. Some other studies also reported results obtained by regression analysis but without reporting its accuracy in a comparison with that of ML [48,49]. Other studies included wide samples of patients [37,38,39] but the accuracy was not lower in studies including small samples [41,50]. Conversely, the largest study was conducted on 2522 patients, which divided into 1522 patients to train an artificial neural network and 1000 patients to test its predictive capacity; it reported an accuracy of only 74%, lower than that obtained on the same samples with conventional statistics (linear regression and cluster analysis) [39].

In TBI patients a similar status is reported, with accuracy ranging from 78% to 98% and no evidence of the best ML algorithm. Indeed, either considering works using mixed or ensemble ML models [30,31,33,34] or that with one single algorithm [32,35], the result is similar: no evidence for a best ML algorithm and no substantial difference with respect to LR approach.

## 5. Conclusions

ML algorithms do not perform better than more traditional regression models in predicting the outcome in TBI or stroke patients. Although ML has been demonstrated to be a powerful tool to capture complex nonlinear function dependencies in several neurological domains [51,52], the state-of-art in TBI and stroke domains do not confirm this advantage. This could be dependent on the type of predictors employed in several studies, such as continuous and categorized (operator-dependent) variables (i.e., clinical scales, radiological metrics). Moreover, ML has demonstrated its value when trained on high-dimensional and complex data extracted from neuroimaging (structural and fMRI), EEG and genetics. Future works are needed to better capture changes in prognosis during intensive care courses extending the current “black-box” or “static” approaches (data extracted from only admission and discharge)” in a new era of mixed dynamic mathematical models [53].

## Figures and Tables

**Figure 1 biomedicines-10-02267-f001:**
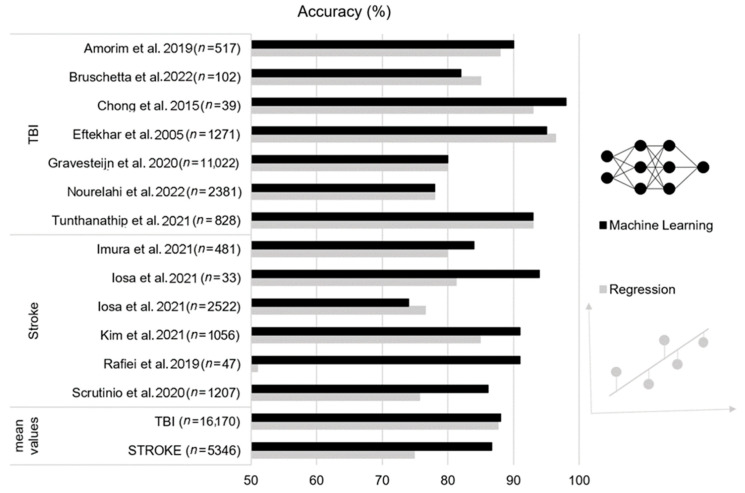
Accuracy (%) of outcome prediction in traumatic brain injury (TBI) and stroke patients for the considered studies (*n:* sample size). Machine learning approach (black bars) versus linear regression (grey bars), [17,18,19,20,21,22,23,24,25,26,27,28,29].

**Table 1 biomedicines-10-02267-t001:** Comparison between classical statistics and machine learning methods.

	Classical Statistics	Machine Learning
**Approach**	Top-down (applied to data)	Bottom-up (extracted by data)
**Model**	Hypothesized by the researcher	Auto-defined
**Power of analysis**	Medium	Usually High
**Accuracy**	Medium	It could be superior or inferior to that of classical statistics
**Reliability**	The same data always provide the same results	The results are affected by the initialization of parameters
**Type of relationships among variables**	Often Linear, in general not complex	Complex relationships
**Interpretability**	Simple	More complex

**Table 2 biomedicines-10-02267-t002:** Comparison between Machine Learning and linear regression approaches in traumatic brain injury patients to predict outcome at discharge.

TBI PATIENTS						
Authors	Algorithms	Sample (*n*°)	Data Type	Outcome	Accuracy *Regression vs. ML Models*	Best Features Extracted
Nourelahi et al. [31]	Logistic Regression Random Forest Support Vector Machine	2381	*Parameters measured at admission:* Age Sex Rotterdam index Blood sugar level Pupil reactivity Coagulation measures prothrombin time-international normalized ratio GCS motor response Systolic blood pressure	Binary outcome based on GOS-e: “favorable” or “unfavorable”	78%/78%	Age GCS motor response Pupil Reactivity Blood sugar level
Tunthanathip et al. [33]	Logistic Regression Support Vector Machine Neural Networks Random Forest Naïve Bayes k-Nearest Neighbor	828	*Baseline and Clinical Characteristics:* Age Gender Mechanism of Injury Association of Injuries Comorbidities GCS score Pupillary light Reflex *Imaging Characteristics:* Skull Fracture Intracranial Injuries Basal Cistern Midline Shift Surgical Treatment	King’s Outcome Scale for Childhood Head Injury	93%/93%	GCS Hypotension Pupillary light Reflex Subarachnoid Hemorrhage
Bruschetta et al. [29]	Logistic Regression Support Vector Machine k-Nearest Neighbors Naïve Bayes Decision Tree	102	Age Sex Marshall Score Entry Diagnosis CRS-R Rancho Los Amigos Levels of Cognitive Functioning Scale Disability Rating Scale ERBI A and B	GOS-e	85%/82%	2 classes: Age CRS-R ERBI A-B 4 classes: Age Sex Entry Diagnosis
Amorim et al. [34]	Generalized Linear model Random Forest Neural Network Decision Tree Boosting Partial Least Square Multivariate Adaptive Regression Splines Naïve Bayes	517	Gender Age Level of pupil reactivity at admission Prehospital GCS GCS at admission GCS motor score Hypoxia Hypotension. Midline shift bigger than 5 mm Brain herniation detected on CT Subarachnoid hemorrhage Epidural hemorrhage Subdural hemorrhage Intracerebral hemorrhage Trauma severity Prothrombin time Partial thromboplastin time	Death within 14 days	88%/90% (Best Model: Naïve Bayes)	GCS at admission Age Prehospital GCS Partial thromboplastin time
Gravesteijn et al. [30]	Logistic Regression Lasso Regression Ridge Regression Support Vector Machine Random Forest Gradient Boosting Machine Artificial Neural Networks	11022	Age Hypoxia Hypotension Marshall CT class Traumatic Subarachnoid Hemorrhage Epidural Hematoma Glucose Hemoglobin GCS motor Pupil GOS	6 months mortalityand unfavorable outcome (GOS < 3, or GOS-e < 5).	80%/80%	N.R.
Eftekhar et al. [32]	Logistic Regression Artificial Neural Networks	1271	GCS Sex Tracheal intubation status Age Systolic blood pressure Respiratory rate Pulse rate Injury Severity Score	Mortality	96.37%/95.09%	N.R.
Chong et al. [35]	Logistic Regression Neural Network	39 children with TBI	*For both methods:* Involvement in road traffic accident History of loss of consciousness Vomiting Signs of base of skull fracture *Only for Neural Network:* Presence of seizure Confusion Clinical signs of skull fracture.	CT scan	93%/98%	Involvement in road traffic accident History of loss of consciousness Vomiting Signs of base of skull fracture Presence of seizure Confusion Clinical signs of skull fracture.

Legend: GCS = Glasgow Coma Scale (GCS), Coma Recovery Scale-revised (CRS-r), Glasgow Outcome Scale-Extended (GOS-e), Early Rehabilitation Barthel Index (ERBI), CT = Computed Tomography, N.R. = not reported.

**Table 3 biomedicines-10-02267-t003:** Comparison between Machine Learning and linear regression approaches in stroke patients to predict outcome at discharge.

Stroke Patients						
Authors	Algorithms	Sample (*n*°)	Data Type	Outcome	Accuracy *Regression* *vs. ML Models*	Best Features Extracted
Rafiei et al. [36]	Enhanced probabilistic neural network General Linear Model	47	Demographical data Stroke-related data Wolf Motor Function Test performance time, fine motor score, gross motor score Touch sensation (Semmes-Weinstein Monofilament Test) Cognitive function (Montreal Cognitive Assessment) Pretreatment daily arm use (MAL).	Multidimensional assessment (Motor Activity Log, Wolf Motor Function Test, Semmes-Weinstein Monofilament Test of touch threshold, and Montreal Cognitive Assessment).	40–51%/85–91%	Sensation Wolf Motor Function Test Gross Motor Score
Scrutinio et al. [37]	Random Forest ADA-Boost and gradient boosting	1207	Demographical data Stroke-related data Functional Independence Measure cognitive and motor Laboratory findings	Death	75.7%/86.1%	Age Severity Time from stroke Functional Independence Measure
Kim et al. [38]	Deep Neural Network Random Forest	1056	Age Type of stroke Medical Research Council scale scores Modified Brunnstrom classification score Functional Ambulation Category score Presence of motor evoked potentials	Modified Brunnstrom classification and Functional Ambulation Category	84.9%/90% (Deep Neural Network 90%),87–91% (Random Forest)	Presence of motor evoked potentials
Iosa et al. [39]	Artificial Neural Network	2522	Demographical data Stroke related data Bamford Classification Clinical assessment of deficits	Barthel Index	76.6%/74%	Global aphasia Age Neglect
Iosa et al. [28]	Artificial Neural Network	33	Spatio-temporal gait parameters Trunk kinematic parameters during walking	Return to Work	81.3%/93.9%	Double support phase Trunk rotation range
Imura et al. [40]	Support Vector Machine k-Nearest Neighbors Random Forest Decision Tree	481	Demographic data Stroke related data Brunnstrom recovery stage Functional independence measure scores	Home discharge	79.9%/84.0% (k-Nearest Neighbors), 82.6% (Support Vector Machine), 79.9% (Decision Tree), 79.9% (Latent Dirichlet Allocation), 81.9% (Random Forest)	N.R.

Legend: GCS = Glasgow Coma Scale (GCS), Coma Recovery Scale-revised (CRS-r), Glasgow Outcome Scale-Extended (GOS-e), Early Rehabilitation Barthel Index (ERBI), CT = Computed Tomography, N.R. = no reported.

## Data Availability

Not applicable.

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
