# Peer review of "Predicting Outcome in Patients with Brain Injury: Differences between Machine Learning versus Conventional Statistics"

_biomedicines, 2022, doi:10.3390/biomedicines10092267_

Round 1

Reviewer 1 Report

The review is about an interesting topic to compare the usefulness of the more conventional statistical methods with the more recent machine learning algorhythms. Although the major conclusion is negative that is machine learning methods do not seem to be superior, the review can be of interest to clinicians assessing outcome in brain injury patients.

The text is quite comprehensive, but it has some shortcomings that should be addressed before publication.

Section 2 describes the main branches of machine learning, however, it does not describe at all the different machine learning techniques that will be subsequently poured onto the reader in the following section without any explanation or distinction. Although detailed mathematical description of these techniques would clearly exceed the scope of this review, it would be still important to briefly introduce the more than half dozen techniques - what are the major differences among these concerning their applicability in prediction of brain injury outcome? A clearly stylistic, but nevertheless very important issue is the use of the abbreviations in the text and the tables. In theory, all abbreviations should be introduced at first mention in the text AND also in each table (the table should be legible without reference to the text). This does not apply to the manuscript in present for hindering its clarity. It would be advisable not to use abbreviations for those terms that do not occur more than 3 times in either the text or the tables.

Author Response

We would like to express our appreciation for the reviewers’ comments. We feel that our manuscript is strongly improved by incorporating their suggestions. In the attached reply to the reviewers, we outlined our responses to each of their comments.

REVIEWER N°1

The review is about an interesting topic to compare the usefulness of the more conventional statistical methods with the more recent machine learning algorhythms. Although the major conclusion is negative that is machine learning methods do not seem to be superior, the review can be of interest to clinicians assessing outcome in brain injury patients.

The text is quite comprehensive, but it has some shortcomings that should be addressed before publication.

  1. Section 2 describes the main branches of machine learning, however, it does not describe at all the different machine learning techniques that will be subsequently poured onto the reader in the following section without any explanation or distinction. Although detailed mathematical description of these techniques would clearly exceed the scope of this review, it would be still important to briefly introduce the more than half dozen techniques - what are the major differences among these concerning their applicability in prediction of brain injury outcome?

REPLY: AS required by this reviewer we re-formulated this section:

Among the wide number of possible machine learning algorithms, there are some conventional techniques that are considered the gold standard for classification problems and that have been employed in the studies presented in this review:

  • Logistic Regression (LR): the simplest among classification techniques, it is mainly used for binary problems. Making the assumption of linear decision boundaries, LR works by applying a logistic function in order to model a dichotomous variable of output. (Tolles & Meurer, 2016)

Where x is the input variable.

This oversimplified model allows low training time and poor possibility of overfitting but at the same ,time it may carry to underfitting for complex datasets. For these reasons logistic regression is suitable for simple clinical datasets such as that related to patients with brain injuries.

Ridge Regression and Lasso Regression are distinguished from Ordinary Least Squares Regression because of their intent to shrink predictors by imposing a penalty on the size of the coefficients. So, they are particularly useful in the case of big data problems.

  • Generalized Linear Models are an extension of linear models where data normality is no longer required because predictions distribution is transformed in a linear combination of input variables X throughout the inverse link function h:

Moreover, the unit deviance d of the productive exponential dispersion model (EDM)

is used instead of the squared loss function.

  • Support Vector Machine (SVM) [20]: it applies a kernel function with the aim to map available data into a higher dimensional feature space where they can be easier separated by an optimal classification hyperplane.
  • k-Nearest Neighbors (k-NN) [21]: it assigns the class of each instance computing the majority voting among its k nearest neighbors. This approach is very simple but requires some not trivial choices such as the number of k and the distance metric. Standardized Euclidean distance is one of the most used because neighbors are weighted by the inverse of their distance:

Where q is the query instance,  is the i-th observation of the sample and  is the standard deviation.

  • Naïve Bayes (NB) [23]: based on the Bayes’ Theorem, it computes for each instance the class with the highest probability of applying density estimation and assuming independence of predictors.
  • Decision Tree (DT) [17]: a tree-like model that works performing for each instance a sequence of cascading tests from the root node to the leaf node. Each internal node is a test on a specific variable, each branch descending from that node is one of the possible outcomes of the test and each leaf node corresponds to a class label. In particular, at each node the function Information Gain is maximized to select the best split variable:

Where I represents the information needed to classify the instance and it is given by the entropy measure

With p(c) equal to the proportion of examples of class c

And Ires is the residual information needed after the selection of variable A:

A common technique employed to enhance models’ robustness and generalizability is the ensemble method that combines predictions of many base estimators. The aggregation can be done with the Bootstrap Aggregation technique (Bagging) applying the average among several trees trained on a subset of the original dataset (such as in the case of Random Forests) or with the Boosting technique applying the single estimators sequentially giving  higher importance to samples that were incorrectly classified from previous trees (like in AdaBoost algorithm)

  • Artificial Neural Networks (ANNs): are a group of machine learning algorithms inspired by the way the human brain performs a particular learning task.

In particular, neural networks consist of simple computational units called neurons, connected by links representing synapses, that are characterized by weights used to store information during the training phase.

A standard NN architecture is composed of an input layer whose neurons represent input variables {xi| x1, x2, ….., xm}, a certain number of hidden layers for intermediate calculations and the output layer that converts received values in outputs.

Each internal node transforms values from the previous layer using a weighted linear summation (u=w1x1 + w2x2 + …. + wmxm) , followed by a non-linear activation function (y=ϕ(u+b)) such as step, sign, sigmoid or hyperbolic tan functions. The learning process is performed throughout the backpropagation algorithm which computes the error term from the output layer and then back propagates this term to previous layers updating weights. This process is repeated until a certain stop criterion or a certain number of epochs are reached.

  1. A clearly stylistic, but nevertheless very important issue is the use of the abbreviations in the text and the tables. In theory, all abbreviations should be introduced at first mention in the text AND also in each table (the table should be legible without reference to the text). This does not apply to the manuscript in present for hindering its clarity. It would be advisable not to use abbreviations for those terms that do not occur more than 3 times in either the text or the tables.

REPLY: Done

Reviewer 2 Report

The authors discussed the main differences between ML techniques and traditional statistics (such as logistic regression, LR) applied for predicting outcomes in patients with stroke and traumatic brain injury (TBI). Thirteen papers directly addressing the different performance among ML and LR methods were included in this review. There are several major problems:

1.      In Table 1: Comparison between classical statistics and machine learning methods, the comparison was not accurate and should be explained in much more detail. For example, in the row of Reliability, Classical Statistics: The same data always provide the same results; Machine Learning: The results could change reiterating the analysis even if on the same data. I think most machine learning methods will get the same results for the same data as well.

2.      Table 2: Machine Learning algorithms employed in brain injury studies to predict the outcome at discharge, were too long. The authors need to better summarize the results. It can be divided into two tables.

3.      The authors only listed several studies and did not do too much further analysis.

4.      Can the data from these previous studies be combined to compile a benchmark dataset?

5.      Are there suggestions on how to choose the analysis method for TBI?

6.      The authors should add some summary figures.

Author Response

We would like to express our appreciation for the reviewers’ comments. We feel that our manuscript is strongly improved by incorporating their suggestions. In the attached reply to the reviewers, we outlined our responses to each of their comments.

REVIEWER N°2

The authors discussed the main differences between ML techniques and traditional statistics (such as logistic regression, LR) applied for predicting outcomes in patients with stroke and traumatic brain injury (TBI). Thirteen papers directly addressing the different performance among ML and LR methods were included in this review. There are several major problems:

    1. In Table 1: Comparison between classical statistics and machine learning methods, the comparison was not accurate and should be explained in much more detail. For example, in the row of Reliability, Classical Statistics: The same data always provide the same results; Machine Learning: The results could change reiterating the analysis even if on the same data. I think most machine learning methods will get the same results for the same data as well.

REPLY: In the row of Reliability of Table 1, we actually wanted to highlight the dependence of machine learning performances on the initialization of model parameters and on the training-test split of the dataset. Clearly, as the reviewer correctly suggested, the results of a well-trained and robust model of machine learning don’t change for the same data except for negligible variations. We changed the following sentence “The result could change reiterating the analysis even if on the same data” with “The results are affected by the initialization of parameters”

    1. Table 2: Machine Learning algorithms employed in brain injury studies to predict the outcome at discharge, were too long. The authors need to better summarize the results. It can be divided into two tables.

REPLY: Table 2 has been splitted

    1. The authors only listed several studies and did not do too much further analysis.
    2. Can the data from these previous studies be combined to compile a benchmark dataset?

REPLY: We would like to thank this reviewer for this suggestion, but as stated in the introduction, this study is aimed at summarizing the first evidence about the impact of machine learning tools in the prediction of outcomes with respect to classical statistical methods. The quantitative evaluation (i.e. benchmark analysis) and comparison among methods and algorithms are outside the main interest of this study. We are interested in describing the main results reached up to now in this field of study, where a direct comparison between traditional and ML methods has been directly tested. Moreover, we are interested in advising clinical and computational scientists to take care in the employment of this challenging method because final results could be affected by methodological choices. We believe that this kind of paper could be very helpful for future research in order to help colleagues in choosing and understanding the best statistical methods. In the past, I wrote a similar paper in the field of Alzheimer’s disease (doi: 10.3389/fnagi.2017.00329), which reached 268 citations in 5 years.

However, in the discussion we now added this sentence

“Qualitative evaluation of results suggested a trend towards better performance of ML algorithms in the stroke patients with respect to LR. However, without a quantitative comparison (i.e., benchmark analysis) a definitive conclusion cannot be drawn.”

    1. Are there suggestions on how to choose the analysis method for TBI?

REPLY: For stroke patients, we strongly suggested the ANN approach, whereas in TBI patients, since the large heterogeneity in best features extracted and the similarity in the statistical performance a definitive suggestion cannot be provided.

    1. The authors should add some summary figures.

REPLY: As suggested by this reviewer a new figure has been added.

Round 2

Reviewer 2 Report

The authors have made significant improvement.